# Target-Specific Nanoparticle Polyplex Down-Regulates Mutant *Kras* to Prevent Pancreatic Carcinogenesis and Halt Tumor Progression

**DOI:** 10.3390/ijms24010752

**Published:** 2023-01-01

**Authors:** Jill P. Smith, Wenqiang Chen, Narayan Shivapurkar, Monica Gerber, Robin D. Tucker, Bhaskar Kallakury, Siva Sai Krishna Dasa, Ruvanthi N. Kularatne, Stephan T. Stern

**Affiliations:** 1Department of Medicine, Georgetown University, Washington, DC 20007, USA; 2Department of Pathology, Georgetown University, Washington, DC 20007, USA; 3Nanotechnology Characterization Laboratory, Cancer Research Technology Program, Frederick National Laboratory for Cancer Research Sponsored by the National Cancer Institute, Frederick, MD 21702, USA

**Keywords:** pancreatic cancer, cholecystokinin receptor, mutant *Kras*, nanoparticles

## Abstract

Survival from pancreatic cancer is poor because most cancers are diagnosed in the late stages and there are no therapies to prevent the progression of precancerous pancreatic intraepithelial neoplasms (PanINs). Inhibiting mutant *KRAS*^G12D^, the primary driver mutation in most human pancreatic cancers, has been challenging. The cholecystokinin-B receptor (CCK-BR) is absent in the normal pancreas but becomes expressed in high grade PanIN lesions and is over-expressed in pancreatic cancer making it a prime target for therapy. We developed a biodegradable nanoparticle polyplex (NP) that binds selectively to the CCK-BR on PanINs and pancreatic cancer to deliver gene therapy. PanIN progression was halted and the pancreas extracellular matrix rendered less carcinogenic in *P48-Cre/LSL-Kras^G12D/+^* mice treated with the CCK-BR targeted NP loaded with siRNA to mutant *Kras*. The targeted NP also slowed proliferation, decreased metastases and improved survival in mice bearing large orthotopic pancreatic tumors. Safety and toxicity studies were performed in immune competent mice after short or long-term exposure and showed no off-target toxicity by histological or biochemical evaluation. Precision therapy with target-specific NPs provides a novel approach to slow progression of advanced pancreatic cancer and also prevents the development of pancreatic cancer in high-risk subjects without toxicity to other tissues.

## 1. Introduction

The incidence of pancreatic cancer is on the rise and will soon become the 2nd leading cause of cancer-related deaths in the USA [1]. With the current chemotherapeutic regimens, the 5-year survival is only about 11% and is only 3% for patients who present with metastatic disease [2]. Reasons contributing to the poor outcome of this malignancy is the inability to diagnose pancreatic cancer in early or precancerous stages [3], the lack of prevention strategies for high risk individuals, its high metastatic potential [4], its relative resistance to chemotherapy [5,6] and the dense fibrotic tumor microenvironment [7]. Pancreatic cancer is considered a “cold tumor” and lacks CD8+ tumor infiltrating lymphocytes, and using immune checkpoint antibodies has thus far largely failed in this malignancy [8]; therefore, chemotherapy remains the mainstay treatment for advanced pancreatic cancer. Monotherapy is no longer recommended in treating advanced pancreatic cancer and combination therapeutic regimens have become the standard of care in order to improve survival [9]. Aggressive modified chemotherapy regimens have improved the overall survival in subjects R0 or R1 resections [10,11]. Yet, even with combination therapy, the 5 year survival of pancreatic cancer remains low and new strategies are needed to overcome the above obstacles. Unfortunately, most treated with chemotherapy experience side effects from the nonspecific and off-target toxicity of these agents [12].

One new approach being explored to change the outcome of advanced pancreatic cancer includes precision medicine with nanomedicines [13,14]. The first nanomedicine applied to treating pancreatic cancer was albumin nanoparticle (nab)-paclitaxel [15], which improved the tolerability and efficacy of paclitaxel when combine with gemcitabine [16]. Nab-paclitaxel has also been modified with embedded tumor necrosis factor-related apoptosis-inducing ligand to improve delivery through the dense fibrosis of the pancreatic cancer microenvironment [17]. Another strategy has been to target pancreatic cell surface receptors in order to improve uptake, efficacy and to limit off-site toxicity. Some nanomedicines rely upon the enhanced permeability and retention effect (EPR) for passive uptake into tumors. Alternatively, active targeting by modification of nanomedicines to render them target-specific has increased the efficacy and reduced the toxicity of these agents [18,19]. Another advantage being employed in nanomedicine is the ability to safely deliver small interfering RNAs (siRNAs) to downregulate genes regulating proliferation or ‘cancer driver genes’ in vivo [14].

We designed a biodegradable nanoparticle polyplex (NP) micelle that selectively targets the cholecystokinin-B receptor (CCK-BR) [20], growth factor receptor that is markedly over-expressed in pancreatic cancer and not detected in the normal pancreas [21,22]. The NP is targeted to the CCK-BR by conjugation of a gastrin decapeptide ligand to the polyethylene glycol surface of the micelle; gastrin is the endogenous peptide hormone ligand of the CCK-BR. Using this platform [20], we previously showed that the CCK-BR targeted NP could deliver siRNA to orthotopic pancreatic tumors in mice to downregulate gastrin, as gastrin peptide stimulates growth of pancreatic cancer [23,24] whereas, the untargeted NP had no effect. In this prior study [20], the CCK-BR targeted NP not only decreased growth of established human pancreatic tumors in athymic nude mice, but also prevented metastases. Since CCK-BRs also become expressed in precancerous pancreatic intraepithelial neoplasia called PanINs [25], we modified our CCK-BR targeted NP with fluorophores to render it as an imaging tool. The fluorophore conjugated NPs directed to the CCK-BR selectively identified high grade PanIN lesions in *P48-Cre/LSL-Kras^G12D/+^* mice in vivo [26]. These prior studies support that the CCK-BR can serve as a potential target for other therapeutics and diagnostics, i.e., a theranostic agent.

Mutant *KRAS* (*muKRAS*) has been identified as a driver gene in human pancreatic cancer and is identified in about 90% of those with this tumor [27]. Until recently, *KRAS*-driven tumors were thought to be ‘undruggable’ [28]. With the discovery and recent approval of allele-specific covalent inhibitors targeting *KRAS*^G12C^ in non-small-cell lung cancer [29], there remains hope that therapies may also be developed to target the common *KRAS*^G12D^ mutation of pancreatic cancer. In the current investigation, we examined the role of the targeted NP to prevent pancreatic carcinogenesis in *P48-Cre/*LSL-*Kras*^G12D/+^ mutant *Kras* mice. We further studied whether the CCK-BR targeted NP could deliver the murine *Kras*^G12D^ siRNA payload to an immune competent mouse model with an established orthotopic syngeneic pancreatic tumor. Next, we examined the safety and efficacy of the CCK-BR targeted NP to deliver mutant *KRAS*
^G12D^ siRNA alone, gastrin (*GAST*) siRNA alone, or both siRNAs in combination in mice bearing orthotopic human pancreatic tumors with a heavy tumor burden to represent the advanced stage in which most human subjects present at diagnosis. Safety and off-target toxicity were also studied after short and long-term NP therapy in immune competent mice.

## 2. Results

### 2.1. Targeted NPs with siRNA to Mutant Kras Prevent PanIN Progression

Currently there are no effective therapies to prevent pancreatic carcinogenesis especially in those at high risk for developing pancreatic cancer. In this investigation, chronic treatment of *P48-Cre/LSL-Kras^G12D^* (KC) mice with CCK-BR targeted NPs loaded with siRNA to *muKras* halted progression of precancerous PanIN lesions. A representative Hemotoxylin & Eosin (H&E) image from a control, untreated *P48-Cre/LSL-Kras^G12D^* mouse pancreas at 6 months of age shows that the normal pancreatic architecture is replaced with high grade PanIN lesions and extensive fibrosis in the extracellular matrix (Figure 1A). Similar histologic findings were found in an age-matched *P48-Cre/LSL-Kras^G12D^* mouse treated with the targeted NP loaded with a scrambled siRNA for *muKras* (Figure 1B). In contrast, a representative image from a 6-month-old *P48-Cre/LSL-Kras^G12D^* mouse treated with the targeted NP loaded with a siRNA for *muKras* reveals very few PanIN lesions and little fibrosis with the preservation of normal acinar cells and structure of the pancreas (Figure 1C). The stained slides were scanned and the number and grade of PanIN lesions counted. Figure 1D shows that the majority of PanINs in the pancreas of control mice and in the pancreas of mice treated with the targeted NP loaded with the scrambled siRNA were high grade, or PanIN-3 grade lesions. In contrast, the analysis of the pancreata from mice treated with the NP loaded with the *muKras* siRNA payload revealed a shift to lower grade PanIN lesions with some early PanIN-1 lesions identified. The difference between the number of high grade PanINs in the *muKras* siRNA-treated group was significantly less than the control and scrambled groups (*p* = 0.0003). The area of the pancreas with normal acinar cells was analyzed with the Aperio Image software for each image and results confirmed that the area of normal pancreas parenchyma histologically was significantly increased or preserved in mice treated with the targeted NP with *muKras* siRNA (Figure 1E). Likewise, the mean number of high grade PanIN-3 lesions per pancreas tissue section was significantly decreased in the *muKras* siRNA-treated mice compared the controls and scrambled siRNA treated mice (Figure 1F). In order to confirm that the reason for the change in PanIN grade and pancreatic histology was due to our therapy, *muKras* mRNA expression in the mouse pancreas was analyzed by qRT-PCR. We found a significant decrease in *muKras* mRNA expression in the pancreas tissue of mice treated with the NP loaded with *muKras* siRNA confirming that the NP successfully delivered its payload (Figure 1G).

### 2.2. Targeted NPs Loaded with Mutant Kras siRNA Decrease Fibrosis in the Pancreas Extracellular Matrix

During pancreatic carcinogenesis, the normal parenchyma of the pancreas is replaced with the precancerous PanIN lesions and dense fibrosis in the pancreatic extracellular matrix (ECM) or tumor microenvironment (TME). This characteristic desmoplastic reaction found in the pancreas of the KC mice and also in human pancreatic cancer is due to the activation of pancreatic stellate cells and fibroblasts [30]. The pancreas from control KC mice (Figure 2A) and from the KC mice treated with CCK-BR targeted NPs loaded with a scrambled siRNA (Figure 2B) exhibited marked intra-pancreatic fibrosis as shown in the representative images of sections stained with Masson’s trichrome. KC mice treated for 4 months with targeted NPs loaded with *muKras* siRNA not only had fewer high grade PanIN lesions and preservation of pancreatic acinar cells, but also exhibited significantly less fibrosis within the pancreas extracellular matrix (Figure 2C). The amount of fibrosis was analyzed by integrative density and demonstrated to be about half the amount as that in the pancreas of control and scrambled siRNA treated mice (Figure 2D; *p* = 0.0015). Confirmation of collagenous fibrosis staining in the pancreas microenvironment was performed with Sirius red staining. Similar to the Masson’s trichrome stains, pancreata of control mice (E) and scrambled siRNA-treated mice (F) show significant staining consistent with increase fibrosis. In contrast, pancreata of mice treated with the *muKras* siRNA loaded NP had less Sirius red staining (G). Densitometry of images from each treatment group (H) revealed less fibrosis in pancreas of the *muKras*-treated mice (*p* < 0.0001).

### 2.3. Targeted NPs Loaded with muKras siRNA Alter Immune Cell Signature by Decreasing M2-Polarized Macrophages

The immune cell signature of the pancreas ECM was markedly changed in the *muKras* siRNA NP-treated mice as demonstrated by a decreased number of M2-polarized arginase positive macrophages. The immunosuppressive M2-polarized macrophages are abundant in the pancreatic tissue of control KC mice (Figure 3A) and also in the pancreas of mice treated with the NP loaded with scrambled siRNA (Figure 3B). However, fewer M2-polarized arginase positive macrophages were found in the pancreas sections of mice treated with the NPs loaded with the *muKras* siRNA (Figure 3C). The number of M2 polarized macrophages were quantified and significantly less in the pancreas of mice treated with the targeted NP loaded with *muKras* siRNA (*p* = 0.02; Figure 3D). These changes in the pancreas ECM rendered the environment less carcinogenic. In order to confirm the immunoreactivity was associated with tissue macrophages, we also performed immunoreactivity with a F4/80 that reacts to murine macrophages. An abundance of macrophages are identified in the pancreas microenvironment of control mice (Figure 3E) and also in the pancreas of scrambled siRNA NP-treated mice (Figure 3F). Significantly fewer F4/80+ tissue macrophages were found in the pancreas of mice treated with the *muKras* siRNA (Figure 3G). Quantitative analysis of macrophage density confirmed less macrophages were present in the pancreas of mice treated with the *muKras* siRNA NPs (Figure 3H; *p* < 0.0001).

### 2.4. CCK-BR Targeted NP Improves Survival and Decreases Metastases of Orthotopic Human Pancreatic Cancer Tumors in Mice

In these series of experiments, we examined the effects of CCK-BR targeted NPs to improve survival and decrease metastases in athymic nude mice bearing orthotopic human pancreatic cancer. In order to simulate a scenario more representative of human subjects presenting with pancreatic cancer, we allowed the tumors to grow up to 5 weeks (PANC-1) before initiating therapy such that the mice had heavy tumor burdens and probable metastatic disease when treatment was started. Two separate experiments were performed using human AsPC-1 and human PANC-1 pancreatic cancers orthotopically implanted to the mouse pancreas. Two different doses of NPs were used with AsPC-1 mice receiving 240nM and PANC-1 mice receiving 480nM. In both experiments, confirmation of established tumors and size was made by abdominal ultrasound prior to initiation of therapy. We previously showed that CCK-BR targeted NPs delivering a gastrin (*GAST*) siRNA payload could prevent pancreatic cancer metastases in mice bearing PANC-1 and BxPC-3 tumors when treatment with nanoparticles was initiated one week after tumor cell inoculation. We wanted to validate the efficacy of the *GAST* siRNA loaded nanoparticle in AsPC-1 cells, a cancer that produces >5 fold more gastrin than PANC-1 cells [31]. The goal of these experiments was to test the effectiveness of the siRNA-loaded nanoparticle in preventing metastases in a mouse with a heavy tumor burden and also to examine whether combination treatment with *GAST* siRNA and *muKRAS* siRNA NPs could be co-administered and enhance the efficacy.

At euthanasia, the organs of each mouse were dissected, fixed, paraffin embedded and stained with H&E and metastases counted and confirmed histologically by the team pathologist in a blinded fashion. The mean number of metastases per mouse in the AsPC-1 tumor bearing mice is shown in Figure 4A according to the treatment group. PBS-treated and scrambled siRNA NP-treated mice had a similar number of metastases. Mice treated with the targeted NP that was loaded with *muKRAS* siRNA had 50% fewer metastases per mouse, but the difference did not reach statistical significance (*p* = 0.0601) at the dose used (240 nM). *GAST* siRNA loaded target-specific NPs decreased the mean number of metastases by >61% (*p* = 0.0149). The number of metastases per mouse was also decreased by 68% (*p* = 0.0163). These results show that in the AsPC-1 tumors, that make an abundant amount of gastrin, the CCK-BR-targeted NP loaded with *GAST* siRNA decreases metastases at the lower 240 nM concentration. Although metastases were fewer in mice treated with either the *GAST* or *muKRAS* siRNA monotherapies, further decrease in metastases was not observed when the two therapies were administered in combination. Perhaps providing the siRNA-loaded NPs at a higher concentration or more frequent intervals would have enhanced the combined effect on lowering metastases. Control tumors and tumors of *GAST* siRNA-treated mice were homogenized and analyzed for gastrin peptide by ELISA. Figure 4B shows the high gastrin peptide concentrations in AsPC-1 control tumors compared to the significantly lower level (*p* = 0.0003) of gastrin peptide from the tumors of mice treated with the *GAST* siRNA NPs confirming the NP delivered the siRNA payload.

We also confirmed that the changes observed in metastases were indeed secondary to the siRNAs down regulating their respective mRNAs; hence, RNA was extracted from the tumors and analyzed by qRT-PCR. Figure 4C shows the gastrin mRNA fold change in tumors of *GAST* siRNA NP-treated mice compared to controls. *GAST* siRNA-treated mice exhibited a 61% decrease in gastrin mRNA expression, and an 82% decrease in expression of gastrin mRNA was observed when the combined siRNA NPs were administered (*p* = 0.0002), suggesting the potential for an additive or synergistic effect. Similarly, mRNA expression of *muKRAS* (Figure 4D) was significantly decreased in tumors of mice treated with *muKRAS* siRNA loaded NPs (*p* = 0.0386); however, no further change in *muKRAS* mRNA was found when combined with *GAST* siRNA loaded NPs.

Metastases were confirmed by the team pathologist by histology with H&E staining. Representative histologic metastases are shown in Appendix A confirming the invasiveness of this orthotopic AsPC-1 tumor. Tumor emboli were identified adjacent to the colon mesentery (Figure 4E) in a mouse treated with the NP loaded with siRNA to *muKRAS*. A tumor emboli is shown in a PBS-treated control mouse in a hepatic vessel (Figure 4F), and another liver vessel tumor emboli is shown in a mouse treated with the scrambled control siRNA loaded NP (Figure 4G). Remarkably, there were no tumor emboli found in tissue sections from the mice treated with the *GAST* siRNA-loaded NP. The lack of tumor emboli in this treatment group may in part explain why fewer metastases occurred in *GAST* siRNA-NP treated mice. This experiment confirmed our ability to deliver two siRNA payloads simultaneously.

In the second human pancreatic cancer orthotopic model with PANC-1 tumors, in order to simulate a scenario more representative of human subjects presenting with advanced pancreatic cancer, we allowed the tumors to grow up to 5 weeks before initiating therapy such that the mice had heavy tumor burdens and probable metastatic disease when the treatment was initiated. The mice were imaged by abdominal ultrasound one week after orthotopic tumor cell implantation (Figure 5A) to confirm tumor formation and again after 5 weeks of tumor growth (Figure 5B). These representative ultrasound images confirmed tumor growth and the large tumor burden at the time therapy was started. A Kaplan Meier survival curve (Figure 5C) of untreated control mice compared to mice treated with the CCK-BR targeted NP loaded with *GAST* siRNA (480 nM) shows a significant survival benefit (*p* = 0.02) even in mice with a large tumor burden. Over one month after the last control mouse died, there were still 3 surviving mice out of the 10 mice treated in the *GAST* siRNA targeted NP group. The mean tumor mass of the surviving mice was significantly (53%) less than that of the control mice and the *GAST* siRNA NP-treated mice that died before the experiment was terminated (Figure 5D). Tumors and metastases were confirmed by H&E histology. Overall, the number of metastases in the *GAST* siRNA NP-treated mice was 64% fewer than the control mice (Figure 5E). Of the 3 mice that were surviving when the experiment was terminated, 2 had no metastases and one had a single metastasis to the spleen. This experiment shows that the *GAST* siRNA loaded NP is capable of improving survival and lowering metastases even when the treatment is started late, similar to that of many human subjects diagnosed in late stages with pancreatic cancer.

### 2.5. Targeted NP Therapy with Kras siRNA Improves Survival in an Immune Competent Mouse

In the third orthotopic tumor model, the effect of the CCK-BR targeted NP to deliver siRNA to silence *muKras* and *WT-Kras* in immune competent mice bearing orthotopic syngeneic mT3 murine pancreatic cancer cells was evaluated. The goal of this experiment was to analyze whether the NPs would improve survival and decrease tumor proliferation in an immune competent syngeneic model. Representative images of the Ki67 immunoreactivity in tumors from each treatment group are shown in Figure 6A–C. The proliferation index of the tumors from the control mice and the mice treated with the scrambled siRNA-loaded NP was high with numerous Ki67 immunoreactive positive cells (Figure 6A,B, respectively). The number of Ki67+ cells was visibly less (Figure 6C) in the tumors of mice treated with the NP loaded with *muKras* and *WT-Kras* siRNA implying that proliferation was decreased by this treatment. Quantitative analysis of the number of Ki67 reactive cells revealed 63% fewer immunoreactive cells in the *WT-Kras/muKras* siRNA-NP treated mice compared to controls (Figure 6D; *p* < 0.0001). Although the number of immunoreactive Ki67+ cells were decreased in the tumors of the scrambled siRNA treated mice, this difference was not significant. Since the NPs loaded with the scrambled siRNA are also targeting the CCK-BR, it is possible that transient blockade of this receptor prevented the activation by the endogenous peptides. We noted a similar slight decrease in tumor size of BxPC-3 pancreatic tumors in our prior work [20] when treated with a CCK-BR-targeted NP loaded with a scrambled siRNA. Survival of mice was significantly greater in the mice treated with the NP loaded with siRNA for *muKras* as shown in the Kaplan Meier curve (Figure 6E; *p* = 0.0178). At a time when all the control and scrambled siRNA-treated mice had died, 50% of the mice treated with NPs loaded with siRNA to *WT-Kras/muKras* were alive. The number of metastases per mouse were also decreased by 42% in the mice treated with the NPs loaded with the *WT-Kras/muKras* siRNA compared to control and scrambled siRNA treated mice (*p* = 0.005). This study shows that the CCK-BR targeted NP also can deliver siRNA payload and decrease pancreatic cancer growth and metastases, and improve survival in an immune competent mouse model.

### 2.6. CCK-BR Targeted NPs Exhibit Broad Safety Profile and Lack off Target Toxicity

The major organs (intestine, stomach, liver, kidney, and spleen) were excised at necropsy from the KC mice and examined grossly and microscopically by the team pathologist after staining with H&E to determine if the high dose (480 nM) NP therapy administered chronically over 4 months induced any off-site toxicity or inflammation. Organs from each of the treatment groups (scrambled and *muKRAS* siRNA NPs) were compared to the histology of organs from the untreated age-matched control mice. Representative H&E stained tissues are shown in the Appendix A for the following tissues: mouse intestine (Appendix A); stomach (Appendix A); liver (Appendix A); kidney (Appendix A); and spleen (Appendix A). No inflammation or abnormality was observed in any of the organs including the stomach, an organ that also has CCK-B receptors.

Mouse serum that was collected at the time of euthanasia was assayed for biochemical changes. Routine chemistry, liver transaminases, and amylase values from wild type C57BL/6 mice with short term (4 weeks) exposure and *P48-Cre/LSL-Kras^G12D^* (KC) mice with chronic (14 weeks) exposure to CCK-BR targeted NPs loaded with siRNAs are shown in Table 1. No significant changes in serum chemistries were noted between control mice and NP-treated mice. The serum amylase levels were higher in the KC mice compared to the wild type mice and this most likely was due to the inflammation induced by *muKras* in the mouse pancreas. There was no statistical difference between the serum amylase levels of untreated KC mice and those treated chronically with NPs.

## 3. Discussion

In this investigation, we developed a novel nanovehicle to safely deliver and silence *muKras* in vivo resulting in inhibition of growth and metastases of pancreatic cancer in murine models. Others have used methods such as electroporation [32] to effectively deliver siRNA to *muKras* to tumors in mouse models, but this technique is not practical for translational therapies for human subject. DOPC (1,2-dioleoyl-sn-glycero-3- phosphatidylcholine) nanoliposomes loaded with *Kras* siRNAs have also been used in animal models of lung and colon cancer with success in decreasing metastases, but this platform did not selectively target a codon mutation [33]. In most human pancreatic cancers a transition in the 12th codon of *KRAS* occurs with a change from GGU in the sense strand of the wild type *KRAS* mRNA to GAU in the mutant sense mRNA. This minor change, however, leads to significant phenotypic alterations in the pancreas and drives carcinogenesis. Such a small alteration renders the gene more challenging to target and difficult to confirm effective knockdown of the target. We had designed several primers to confirm efficient knockdown of mutant *KRAS* and found that the primer with the altered codon placed at the 5′ position resulted in effective detection by qRT-PCR of the down-regulation by qRT-PCR. Indeed, we confirmed that the *muKRAS* mRNA was down-regulated in the pancreatic tumors; hence, the targeting polyplex vehicle utilized was effective.

Obviously, the best strategies to improve survival from cancer involve prevention. In the *P48-Cre/*LSL-*Kras*^G12D^ mouse model that spontaneously develops pancreatic cancer through *Kras*-stimulated progression of PanIN lesions [34], we showed that treatment with the NP loaded with *muKras* siRNA significantly halted PanIN progression and altered the fibrosis of the pancreas extracellular matrix thus preventing cancer. High grade or PanIN-3 lesions are considered carcinoma in situ [35] and those with a strong family history of pancreatic cancer often have multi-focal precursor neoplastic PanIN-3 lesions [36]. These subjects could potentially benefit from strategies to prevent the progression of PanINs to cancer. Unfortunately, the majority of those diagnosed with pancreatic cancer has no family history and are considered sporadic in occurrence. Using this same polyplex nanoparticle platform that targets the CCK-B receptor, we are developing an imaging tool that can detect PanIN-3 lesions [26]; such a method could facilitate early detection of high grade PanINs lesions that would be amenable for therapy.

We demonstrated important changes in the pancreas extracellular matrix in the KC mice treated chronically (14 weeks) with the targeted NPs loaded with *muKras* siRNA; there was a considerable decrease in pancreatic fibrosis and an alteration in the immune cell signature. CCK receptors have been reported on rodent pancreatic stellate cells [37] and when stimulated, these release collagen to form the fibrosis in the extracellular matrix. We previously showed that a CCK receptor antagonist, proglumide, prevents fibrosis in the pancreas extracellular matrix in *Pdx1-Cre/LSL-Kras^G12D^* transgenic mice [25] and also decreased the fibrosis in models of pancreatic cancer [38,39]. One explanation for the decreased fibrosis observed in the mice treated with the CCK-BR targeted NPs may have been due to blocking activation of these receptors on stellate cells in the pancreas and also decreasing inflammation mediated by Kras.

Macrophages can polarize to be either tumor killing (M1) or tumor promoting (M2) [40] and arginase I (ARGI) expression is associated with M2, tumor-promoting macrophages [41]. The cancer-promoting pancreatic microenvironment has an abundance of M2 polarized [42,43] as was observed in the pancreas in this investigation of the control and scrambled siRNA NP treated KC mice. Macrophages release chemokines and cytokines known to activate pancreatic stellate cells or fibroblasts [44] contributing to the increase inflammation and fibrosis. The decreased numbers of M2-polarized macrophages seen in the pancreas of mice treated with targeted NP loaded with *muKras* reflect a change in the immune cell signature to a less tumorigenic phenotype.

In addition to mutant *KRAS*, other growth factors have been shown to stimulate growth of human pancreatic cancer, including gastrin [24]. In fact, we previously showed that gastrin collaborates with mutant *Kras* to promote PanIN progression and carcinogenesis in the *P48-Cre/LSL-Kras^G12D^* mouse model [45]. In the presence of gastrin, signal transduction in the KC mouse pancreas occurs by phosphorylation of HER family members including HER2 and EGFR and activation of RAS mitogenic pathways. This proto-oncogenic MAP-kinase signaling pathway leads to cellular proliferation by a non-canonical pathway that is reversed in the absence of gastrin [45]. The polyplex nanoparticle platform used in this investigation was previously used to deliver *GAST* siRNA and was shown to decrease metastases in mice bearing human pancreatic cancers [20]. In fact, down-regulating gastrin expression appears to be as effective in treating pancreatic cancer as down-regulation of *muKRAS*. When both siRNAs were administered in combination, no significant additive or synergistic effect was observed at the doses used.

Mice bearing heavy tumor burdens were purposely studied in this investigation to represent models that parallel the majority of human subjects at the time of diagnosis. Only 10–15% of those with pancreatic cancer have resectable disease at the time of diagnosis and approximately 40–45% of subjects already have metastases [46]. Even at this late stage of disease, we showed that treatment with targeted NPs loaded with siRNA to *GAST* or *muKRAS* decreased the number of metastases and prolonged survival. A unique feature observed in this study was the histologic absence of tumor emboli noted in the mice treated with the *GAST* siRNA loaded NP, while present in the mice treated with the other loaded NPs. We have also shown that inhibition of the gastrin: CCK receptor signaling pathway with a receptor antagonist, proglumide, blocked metastases by decreasing tumoral genes associated with epithelial to mesenchymal transition (EMT) [39]. CCK receptor antagonists also decrease tumor emboli by decreasing the expression of VEGF-A [47]. A cancer vaccine that induces neutralizing antibodies to gastrin was shown to improve survival in a murine model of pancreatic cancer and reduce metastases by decreasing components of the metastases cascade and EMT, including protein expression of paxillin, β-catenin, and MMP-7 [48]. These mechanisms most likely explain the findings in this investigation of decreased metastases and tumor emboli in the murine models treated with the *GAST* siRNA loaded NP.

In drug development, assurance of the safety of the new product is crucial. In this investigation, we examined the efficacy of the targeted NP to decrease pancreatic cancer tumor proliferation and improve survival in an immune competent mouse model. One purpose of using a model with an intact immune system is to assure the therapeutic agent is not immunogenic and is not inactivated in an animal with the ability to generate neutralizing antibodies. Although cationic nanoliposomes provide an efficient way to complex with negatively charged siRNAs, these positively charged nanoparticles are highly toxic [49] eliciting dose-dependent toxicity and pulmonary inflammation, hepatotoxicity and a systemic interferon type I response [50]. Safety and off-target toxicity was examined in our study with immune competent mice over a short (4 weeks) or long (4 months) duration of NP administration. No serological abnormalities were noted in chemistries between control and treated mice. Furthermore, histologic sections of the major organs failed to reveal any inflammation or other pathological changes. Since our polyplex NP was designed to selectively target cells that over-express the CCK-B receptor (like PanINs and pancreatic cancer), this form of precision medicine decreases the likelihood of uptake and off-target toxicity in other organs that lack this receptor. In fact, there was a concern in using this target-specific NP since the mouse and human stomach also expresses the CCK-B receptor and our targeted NP could potentially bind to these receptors. However, no histologic alterations were identified in the mouse stomach sections. Even if the CCK-B receptor targeted NP were to bind to the gastric CCK-B receptor, the enterochromaffin-like cells and parietal cells in the stomach body that have low expression of the CCK-B receptor do not express gastrin mRNA or *muKRAS* so adverse outcomes would not be expected. These data are consistent with the lack of gastric tissue uptake observed in previous imaging studies using this CCK-B receptor targeted polyplex [26]. With the recent broad use and safety of mRNA COVID-19 vaccines [51,52], development of other platforms to deliver mRNA therapies or siRNA therapies, as in our research, may hopefully be accelerated.

## 4. Materials and Methods

### 4.1. Characterization of Cell Lines

AsPC-1 and PANC-1 human pancreatic tumors were obtained from the American Type Culture Collection (Rockville, MD, USA). Cells were maintained in the appropriate media: DMEM with 10% FBS for PANC-1 and RPMI-1640 with 10% FBS for AsPC-1 in humidified air with 5% CO_2_. We previously showed that AsPC-1 and PANC-1 cells express CCK-B receptors and make endogenous gastrin [31]. Before inoculation in mice, the cells were tested by IMPACT II PCR Profile (IDEXX BioAnalytics, Columbia, MO, USA) and were negative for all pathogens. The mT3-2D cells (mT3 cells) were a gift from the laboratory of Dr. David Tuveson, Cold Spring Harbor, NY. This murine pancreatic cancer cell line was derived from *Kras+*/^LSL-G12D^; *Trp*53+/^LSL-R172H^; *Pdx-Cre* (KPC) mice and are syngeneic to C57BL/6 mice [53]. These cells were maintained in DMEM media with 10% FBS in humidified air with 5% CO_2_. The mT3 murine cells have also been shown to have CCK-BR and make endogenous gastrin [48].

### 4.2. Nanoparticle Polyplex (NP) Formulation and Characterization

In order to develop the targeted NP, a thiol functionalized polyethylene glycol-block-poly(L-lysine) (SH-PEG-PLL) polymer was synthesized as previously described [20,26]. To render the NP target-specific for the cholecystokinin-B receptor (CCK-BR) we conjugated maleimide functionalized Gastrin-10 peptide (Glu-Glu-Glu-Ala-Tyr-Gly-Trip-Met-Asp-Phe-NH_2_, MW 1426.48 g/mol) (custom synthesis by GenScript USA Inc., Piscataway, NJ, USA) to SH-PEG-PLL via a Michael addition reaction. The resulting Ga-PEG-PLL was extensively purified by dialysis using 7 kDa MWCO dialysis cassettes against 0.1 M sodium phosphate buffer, pH 7.4 with 0.3 M NaCl. The polyplex micelle was prepared by mixing 1 mg/mL of the Ga-PEG-PLL with double stranded siRNA that selectively targeted *GAST*, human *muKRAS*, murine mutant *Kras*, wild-type (WT)-*Kras* or a nonspecific scrambled sequence control (Appendix A). The sequences for the *muKRAS* and the scrambled control had been previously validated by Réjiba et al. [32]. NCBI blast confirms the homology of the sequences for human and mouse.

The self-assembled targeted dual siRNA polyplex NPs were characterized for hydrodynamic size distribution by dynamic light scattering (DLS) using photon correlation spectroscopy with an optical glass round cuvette with a diameter of 5 mm in a 3D LS Spectrometer (LS Instruments, Fribourg, Switzerland), equipped with a HeNe laser at 633 nm (25 °C, 633 nm laser, 90° scattering angle).

### 4.3. In Vivo Animal Models

All animal studies were performed in an ethical fashion and approved by the Institutional Animal Care and Use Committee (IACUC) at Georgetown University. Three experiments included orthotopically grown pancreatic tumors: two were with athymic nude mice (Charles Rivers) with human AsPC-1 and PANC-1 tumors and the third experiment used immune competent C57BL/6 mice (Charles Rivers) with syngeneic murine mT3 tumors orthotopically explanted in the pancreas. For cancer prevention studies and chronic safety studies, *P48-Cre/*LSL-*Kras*^G12D^ male and female littermates were used. For short-term safety studies male and female wild-type C57BL/6 mice were used. A summary of the different animal models and experiments performed is shown in Table 2. Mice were housed in the Comparative Medicine facility at Georgetown University with 5 mice per cage in filter top cages and fed standard chow and water ad libitum. The rooms were on automatic lighting with a 12-h on-off cycle. Immunodeficient athymic nude mice received autoclaved food, bedding and water.

### 4.4. Evaluation of the Effectiveness of Targeted NPs with siRNA to MuKras to Prevent PanIN Progression

Mice were bred in a transgenic colony and genotyped to obtain *P48-Cre/LSL-Kras^G12D^* mice for experimentation. Male (N = 15) and female (N = 10) littermates were used. Starting at 10 weeks of age (a time when PanIN lesions are beginning to develop, mice were randomly divided into three groups: no treatment (control), targeted nanoparticle with scrambled siRNA, or targeted nanoparticle with *muKras* siRNA. The mice received an ip injection of 480 nM of each NP in a volume of 100 μl twice a week for 14 weeks or until the mice reached 24 weeks of age. The mice were ethically euthanized, blood collected for serum chemistries, pancreas and other organs excised and examined histologically for off-target toxicity. A small sample of pancreas was collected for RNA extraction and the remainder of the pancreas was fixed, paraffin embedded and stained with H&E for histologic evaluation and grading of PanIN lesions.

The pancreata were stained with H&E, scanned with the Aperio GT450 machine, and histology compared between the three cohorts. Using the Aperio Image software, the normal acinar area of each pancreas was measured and compared to the area of pancreas replaced with PanIN lesions. Using the annotation feature on the Aperio software, PanIN lesions in each pancreas were counted and categorized as low grade (PanIN-1 and 2), or high grade (PanIN-3) lesions. The mean number of PanINs at each grade was determined per treatment group. Sections of mouse pancreata were also reacted with Masson’s trichrome stain to analyze the extent of intra-pancreatic fibrosis and with Sirius red to evaluate collagen in the pancreas microenvironment. Stained slides were analyzed with Image-J computer software and integrative density of fibrosis quantified.

RNA was extracted from part of the pancreas (Qiagen, Germantown, MD) and subjected to qRT-PCR using a thermal cycler 7300 (Applied Biosystems, Waltham, MA, USA) as above to examine the expression of *muKras* mRNA using selective primers (Appendix A). A dissociation curve analysis of PCR products was carried out to confirm the specificity of amplification. The relative differences between two groups were calculated using ∆∆CT method. *Hprt* was used as the normalizer.

### 4.5. Orthotopic Pancreatic Cancer Experiments

Forty female athymic nude mice were each anesthetized and 10^6^ AsPC-1 human pancreatic cancer cells were injected into the pancreas as we previously described [20]. Two mice died after surgery or as a complication of the anesthetic. The remaining 38 mice were imaged by ultrasound after 7 days and had evidence of visible tumors; therefore, these mice were divided into the 5 following groups with equal size tumors: control mice (N = 7) were treated with PBS (100 μl) three time per week (TIW); the second control group (N = 7) received the targeted NP with a nonselective scrambled siRNA; the third group received a targeted NP with the *GAST* siRNA (N = 8); the forth group received the targeted NP with the *muKRAS* siRNA (N = 8); and the last group received the targeted NPs with a combination of the *GAST* and the *muKRAS* siRNA (N = 8). The NP injections were administered intraperitoneally (ip) TIW in a volume of 100 μl and with a siRNA concentration of 240 nM. Mice were ethically euthanized when they lost 10% of the body weight, developed ascites, or exhibited morbidity. The metastases were counted, tissues dissected, fixed, paraffin embedded and the slides with tissues stained with H&E for histologic confirmation of metastases by the team pathologist. The goal of this study was to determine the efficacy of a NP loaded with either *GAST* siRNA or siRNA for *muKRAS* in decreasing metastases in an aggressive tumor that expresses very high levels of gastrin.

In another experiment, 20 female athymic nude mice received an orthotopic injection of 10^6^ human PANC-1 cells in the pancreas. In this experiment, the tumors were allowed to grow and metastasize for 31 days before initiating therapy with the intention of resembling a scenario similar to that of human subjects who present with pancreatic cancer. After 1 week, the mice were imaged by ultrasound and two mice did not have evidence of tumors and were not included in the investigation. After 31 days, the mice were reimaged by ultrasound to confirm growth of the tumors and the presence of a heavy cancer burden before initiation of the study. The mice were divided into two groups and treated with PBS (N = 8) or targeted-specific NPs carrying *GAST* siRNA 480 nM (N = 10), each in a volume of 100 μl administered ip three times weekly (TIW). As above, mice were treated until they became moribund and then were ethically euthanized, and tumors removed, weighed and metastases counted. The primary goal of this experiment was to determine if treatment with a target-specific NP with *GAST* siRNA could prolong survival when treatment was initiated in a very advanced stage.

The third orthotopic study was performed in immune competent C57BL/6 mice bearing syngeneic murine pancreatic cancers. Tumors were established by injecting 100,000 luciferase tagged mT3 cells into the pancreas of thirty mice. One week after surgery, luciferin (30 mg/mL) (Nanolight Technology, Pinetop, AZ, USA) was administered to mice by an ip injection in a volume of 100 μl, in order to measure tumor size with the IVIS Lumina III In Vivo Optical Imaging System (Perkin Elmer). Mice were divided into three treatment groups (N = 10 mice each) of equal tumor size at baseline. Treatment was initiated on day 9 after surgery with either PBS (Control), targeted NP carrying scrambled siRNA for *Kras*, or targeted NPs with siRNA for both *WT- Kras* and *muKras*. Treatments were given TIW ip in a volume of 100 μl, and NP siRNA concentrations were 240 nM. The goal of the study was to analyze the growth and survival of mice with the administration of NPs containing two *Kras* siRNA (wild-type and *muKras*) compared to controls.

### 4.6. Confirmation of Gastrin Peptide Down-Regulation by ELISA

Tumors from control mice and *GAST* siRNA-treated mice were homogenized in a buffer containing Tri-HCl 50mM with proteases. Homogenates were centrifuged and the supernatant (100 µL) was assayed in duplicate from each sample using an ELISA from Enzo Life Sciences (Cat# 25-0417; Farmingdale, NY, USA). This assay has 100% reactivity with human gastrin and only 2.67% reactivity with CCK. The standard curve was generated with the following concentrations of gastrin: 10,000, 2500, 625, 156.25 and 39.1 pg/mL. After the incubation period the Colorimetric ELISA was read in a plate-reader at 405 nm and unknowns were calculated using MyAssays software.

### 4.7. Confirmation That the siRNA Was Delivered and Down-Regulated the Gene by qRT-PCR

Total RNA (Qiagen, Germantown, MD, USA) was extracted from each of the tumors at necropsy and subjected to qRT-PCR for determination of expression of genes targeted by the siRNAs. The primers designed to analyze the gene expression are shown in Appendix A Synthesis of cDNA was performed using a qScript cDNA Synthesis Kit (Quanta Biosciences, Gaithersburg, MD, USA). Real-time PCR was performed using a Perfecta SYBR Green FastMix ROX kit (Quanta Biosciences, Gaithersburg, MD, USA) with an Applied Biosystems 7300 Real Time PCR System machine to assess the expression of *GAST* and *muKRAS*. Samples were subjected in triplicate to qRT-PCR or 40 cycles at 60^o^ C. GAPDH was used as the normalizer for human tumors.

### 4.8. Immunohistochemistry Evaluation of Proliferation Index and M2-Polarized Macrophages

Tumors from immune competent mice (mT3) were fixed and paraffin embedded. To determine the proliferation index of the tumors, tissue sections (5 µm) were reacted with a rabbit monoclonal antibody for Ki67 (Biocare, cat# CRM325; 1:80). Slides were scanned using an Aperio GT450 machine and images captured with software from Aperio Image Scope. Ki67 staining was analyzed by densitometry with Image-J software corrected for area of tissue examined. Average Ki67+ stains were analyzed from PBS images (N = 30), scrambled siRNA NP-treated tumors (N = 19), and *muKras* & *WT-Kras* siRNA NP-treated tumors (N = 19). Mean Ki67 staining for each group was analyzed and plotted by Prism GraphPad version 9.

Pancreas tissue macrophages were stained for F4/80 epitope by exposing performing heat induced epitope retrieval (HIER) by immersing the 5 µm tissue sections at 37 °C for 14 min in Pronase. Immunohistochemical staining was performed using a horseradish peroxidase labeled polymer from Vector MP-7444, according to manufacturer’s instructions. Briefly, slides were treated with 3% hydrogen peroxide and 2.5% normal goat serum (from Kit) for 20 min each, and exposed to a rat monoclonal primary antibody for F4/80 against mouse, (eBioscience, San Diego, CA, USA) at a titer of 1:35 for overnight at 4 °C. Slides were exposed to the Impress Anti-Rat IgG (mouse adsorbed) labeled polymer for 30 min and DAB chromagen (Dako) for 5 min. Slides were counterstained with Hematoxylin (Sigma, Harris Modified Hematoxylin, St. Louis, MO, USA) at 1/10 dilution for 2 min at RT, blued in 1% ammonium hydroxide for 1 min at room temperature, dehydrated, and mounted with Acrymount. Sections with the omitted primary antibody were used as negative controls. Slides were scanned using an Aperio GT450 machine and images captured with software from Aperio Image Scope.

Pancreas from the KC mice were fixed and embedded and after antigen retrieval as above, 5 µm sections were reacted with rabbit polyclonal arginase antibody (ThermoFisher, Cat# PA5-29645) at a dilution of 1:1800 to detect M2 macrophages. All slides were reacted with HRP-conjugated anti-rabbit secondary antibody (Agilent Cat# K400311-2). Images were taken on an Olympus BX61 microscope with a DP73 camera. The number of immunoreactive cells per slide area was counted with image-J computer software by an assistant blinded to the treatments.

### 4.9. Safety and Toxicity Assessment

Safety and off-target toxicity of the targeted NP were evaluated in immune competent mice. Wild-type male and female C57BL/6 mice were treated with targeted nonspecific NPs (480 nM) ip in a volume of 100 μl twice weekly for 4 weeks (short term) or an equal number with PBS control. After 4 weeks, blood was collected at euthanasia by cardiac stick and evaluated for serum chemistries, including liver profile, electrolytes, calcium and amylase. Five major organs were dissected (stomach, intestine, kidney, liver and spleen) fixed, paraffin embedded and stained with H&E. Histology slides were examined by the team pathologist for inflammation or off target toxicity.

Safety and off-target toxicity were also evaluated after long-term (14 weeks, chronic therapy) in the P48-Cre/LSL-Kras^G12D^ transgenic mice treated twice weekly with 480 nM targeted NPs (scrambled or *muKras*) compared to untreated controls. Similar to the wild-type mice, blood was collected at euthanasia and analyzed for serum chemistry and the 5 major organs were dissected, fixed, embedded and stained with H&E and examined by the team pathologist for off-target toxicity.

### 4.10. Statistical Analysis

All statistical analyses were performed using GraphPad Prism software version 9.1 or Minitab Version 19. Differences between the treatments were performed using two-way ANOVA. When only two groups were compared (i.e., gastrin ELISA assay, student’s *t*-test was done. Significance was set at the 95% confidence interval with *p* < 0.05 considered significant. Survival analysis was performed with a Kaplan Meier Survival Curve of mice. Log-rank analysis was performed comparing controls to each treatment group by applying a Cox proportional hazard model.

## 5. Conclusions

In conclusion, we demonstrated a safe and effective strategy to decrease growth and metastases of pancreatic cancer using CCK-B receptor targeted polyplex NPs. These NPs effectively down-regulated the expression of *muKRAS* and *GAST*; both have been shown to serve as drivers of pancreatic cancer. Since NP delivery platforms have demonstrated safety and ability to scale-up for clinical translation, use of our NP that selectively targets the CCK-B receptor further increases optimism that *muKRAS* can be targeted and survival from pancreatic cancer improved.

## 6. Patents

Georgetown University and the NIH have an issued patent (US11,246,881).

## Figures and Tables

**Figure 1 ijms-24-00752-f001:**
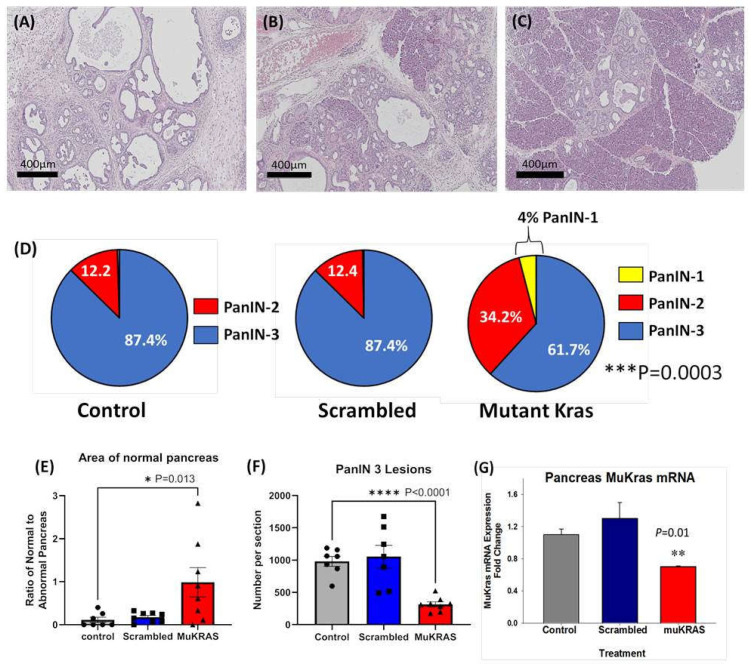
CCK-BR targeted NP loaded with siRNA for *muKras* decreases the number of high grade PanIN lesions in *P48-Cre/LSL-Kras^G12D^* mice. (**A**) H&E stain of a representative 6 month-old control mouse pancreas revealed extensive high grade PanIN lesions and fibrosis replacing the pancreas. (**B**) A representative image of a pancreas from a 6-month old *P48-Cre/LSL-Kras^G12D^* mouse treated with the NP loaded with the scrambled siRNA also shows high grade PanINs and fibrosis. (**C**) A representative H&E stain of a 6 month pancreas of a *P48-Cre/LSL-Kras^G12D^* mouse after 14 weeks of therapy with the target specific NP loaded with *muKras* siRNA shows fewer PanINs and preservation of the normal pancreas histology. (**D**) Pie-shaped diagrams from the three treatment groups shows that 87.4% of the PanINs counted were high grade PanIN-3 lesions in the pancreas of control and scrambled siRNA-treated *P48-Cre/LSL-Kras^G12D^* mice. The number of high grade PanINs are less and the low grade PanINs increased in mice treated with the NP delivering the *muKras* siRNA. This difference was significant (*p* = 0.0003). (**E**) The ratio of normal pancreas area to abnormal pancreas was greater in the pancreas of mice treated with the *muKras* siRNA (*p* = 0.013). (**F**) The absolute number of PanIN-3 lesions per histologic section is shown with significantly less PanIN-3 lesions in the pancreas of the *muKras* siRNA-treated mice (*p* < 0.0001). (**G**) mRNA expression of *muKras* was decreased in the pancreas of mice treated with the NP carrying the *muKras* siRNA payload (*p* = 0.01). * *p* < 0.05; ** *p* < 0.01; *** *p* < 0.005; **** *p* < 0.0001. Individual data points are represented by circles, squares and triangles in the columns.

**Figure 2 ijms-24-00752-f002:**
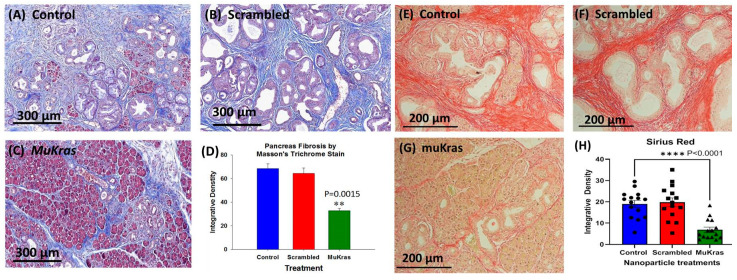
The pancreas extracellular matrix (ECM) shows decreased fibrosis and collagen in mice treated with targeted NP loaded with *muKras* siRNA. (**A**) Representative Masson’s trichrome stain of a control mouse pancreas shows extensive fibrosis and high grade PanIN lesions. (**B**) Representative image from the pancreas of a mouse treated for 4 months with targeted scrambled siRNA also shows extensive pancreatic fibrosis. (**C**) The pancreata from mice treated with the targeted NP loaded with *muKras* siRNA have less fibrosis by Masson’s trichrome stain, fewer PanINs, and preservation of the normal pancreas acinar cells in the parenchyma. (**D**) Quantitative analysis of the amount of fibrosis with computerized integrative density shows significantly less fibrosis in the KC mice treated with targeted NP loaded with *muKras* siRNA (*p* = 0.0015; compared to control & scrambled –treated; two-way ANOVA). (**E**) Representative image of a Sirius red stained pancreas section of control mice and (**F**) in the pancreas of mice treated with the scrambled siRNA loaded NP. (**G**) A decrease in the Sirius red staining was observed in the pancreas of the KC mice treated with the *muKras* loaded NPs. (**H**) Quantitative analysis of the amount of fibrosis with computerized integrative density shows significantly less fibrosis in the KC mice treated with targeted NP loaded with *muKras* siRNA. (*p* < 0.001; compared to control & scrambled-treated; two-way ANOVA). ** *p* < 0.01; **** *p* < 0.0001. Individual data points are represented by circles, squares and triangles in the columns.

**Figure 3 ijms-24-00752-f003:**
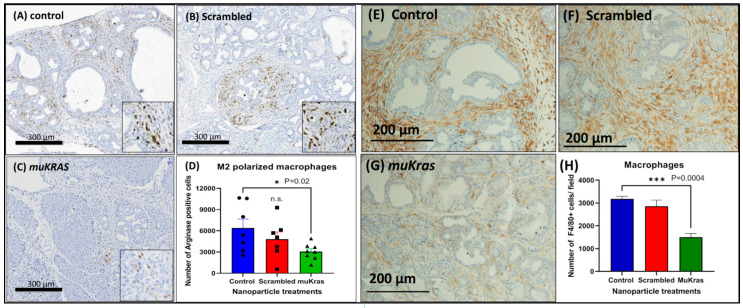
The pancreas microenvironment has fewer immunosuppressive M2-polarized macrophages in mice treated with mu*Kras* siRNA nanoparticles. (**A**) Arginase positive M2-polarized macrophages are seen in the ECM and are abundant in pancreas of control mice and (**B**) in the pancreas of mice treated with the scrambled siRNA loaded NP. (**C**) A decrease in the arginase staining was observed in the pancreas of the KC mice treated with the mutant *Kras* loaded NPs. (**D**) The number of arginase positive macrophages was quantified and the number per high powered field were significantly decreased in the pancreas of the KC mice treated with the mutant *Kras* loaded NP (*p* = 0.02). (**E**) F4/80+ macrophages were abundant in the pancreas of control mice and (**F**) in mice treated with the scrambled siRNA. (**G**) Fewer tissue macrophages were identified in the pancreas of mice treated with the mu*Kras* siRNA NP. (**H**) Quantitative computerized densitometry confirmed fewer macrophages in the pancreas of the mu*Kras* siRNA NP- treated mice (*p* = 0.0004; two-way ANOVA).). *** *p* < 0.005. Individual data points are represented by circles, squares and triangles in the columns.

**Figure 4 ijms-24-00752-f004:**
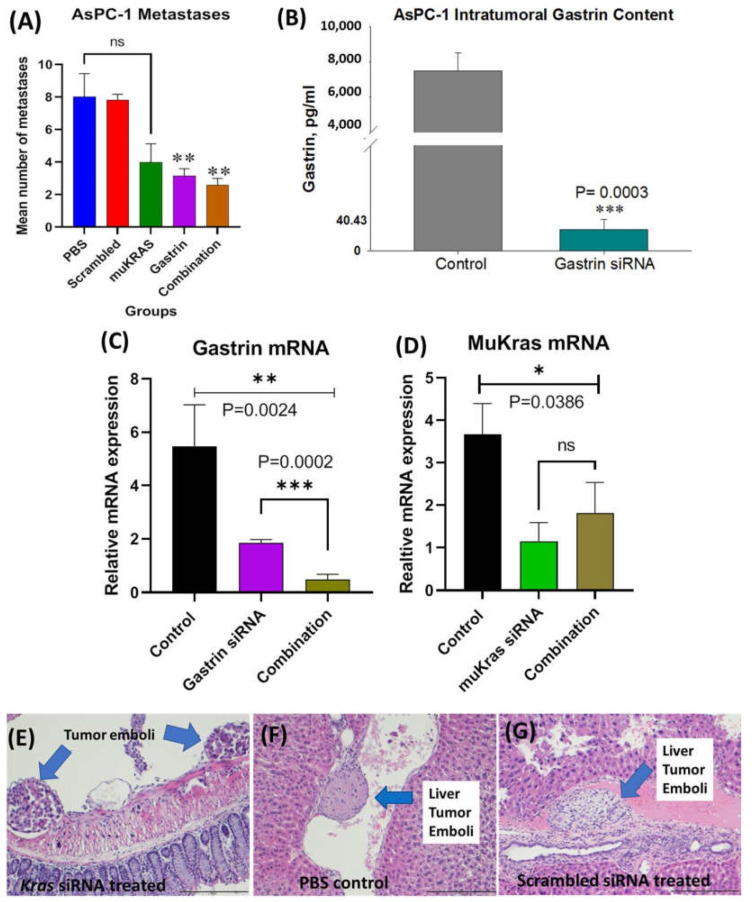
Effects of CCK-BR targeted NP on metastases in human pancreatic cancer in AsPC-1 tumor bearing mice. (**A**) Mean number of metastases per mouse is shown. Mice treated with PBS and the scrambled siRNA NP had the greatest number of metastases. *MuKRAS* siRNA-treated mice had fewer metastases but not statistically significant (*p* < 0.06). However, both groups treated with the NP delivering the siRNA to *GAST* had fewer metastases (*p* < 0.02). (**B**) Gastrin peptide in tumor homogenates measured by ELISA show less gastrin peptide in tumors of mice treated with *GAST* siRNA NPs (*p* = 0.0003). (**C**) Tumors were excised and gastrin mRNA expression was confirmed to be down-regulated by qRT- PCR (*p* = 0.0024). (**D**) *MuKRAS* mRNA expression levels in tumors of mice treated with the *muKRAS* siRNA loaded NP is lower than that of control tumors. (**E**) H&E image of tumor emboli seen adjacent to the colon in a *muKRAS* siRNA treated mouse. (**F**) A tumor emboli is identified in a central vein in a liver section from a PBS-control treated mouse. (**G**) Tumor emboli identified in a hepatic vein of a mouse treated with a scrambled siRNA loaded NP, (scale bar = 200 µm). * *p* < 0.05; ** *p* < 0.01; and *** *p* < 0.005.

**Figure 5 ijms-24-00752-f005:**
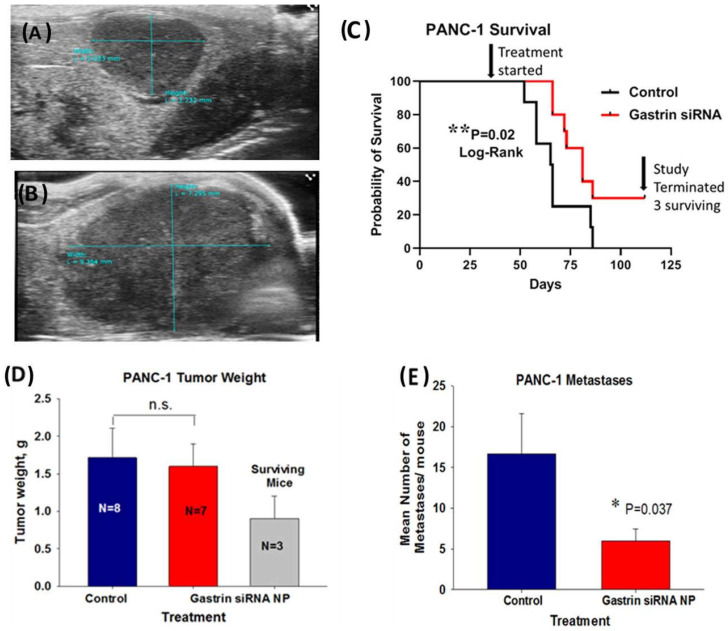
Treatment of mice with a large PANC-1 tumor burden and late disease with a NP loaded with *GAST* siRNA still improves survival and decreases metastases. (**A**) A representative ultrasound image confirms the presence of a PANC-1 tumor one week after inoculation. (**B**) A representative abdominal ultrasound of a mouse is shown 5 weeks after PANC-1 inoculation and at the initiation of therapy with an extensive large tumor. (**C**) Kaplan Meier survival curve shows improved survival benefit (*p* = 0.02) in mice treated with NP loaded with *GAST* siRNA. Top arrow points to the time when the treatment was started. The second arrow on the right points to the time when the study was terminated and to the N = 3 surviving mice in the *GAST* siRNA-treated group over 1 month after the last control mouse had died. (**D**) Final tumor weights in grams are shown for the N = 8 control mice and *GAST* siRNA NP- treated mice that died during the study. The tumor weights of the N = 3 *GAST* siRNA NP treated surviving mice are shown. (**E**) The mean number of metastases counted and confirmed histologically was significantly less in the *GAST* siRNA -treated mice compared to control (*p* = 0.037). * *p* < 0.05; ** *p* = 0.02.

**Figure 6 ijms-24-00752-f006:**
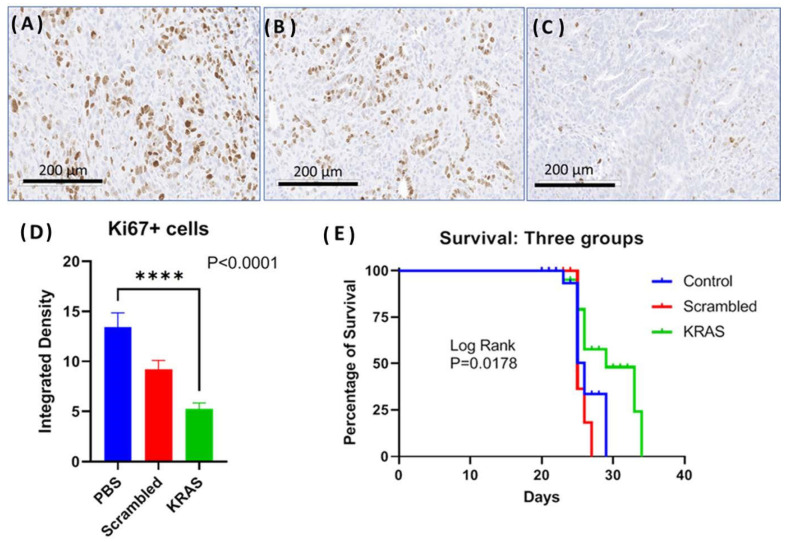
Effects of targeted NP loaded with *muKras* and *WT-Kras* siRNA on growth and survival of mice bearing mT3 syngeneic orthotopic pancreatic tumors. (**A**) Ki67 immunohistochemistry staining of a representative control tumor shows abundant proliferating cells. (**B**) Representative image from a tumor of a scrambled-siRNA NP treated mouse, Ki67 immunohistochemistry staining. (**C**) Representative image from an mT3 tumor of a mouse treated with the NP loaded with siRNA for *muKras* and *WT-Kras* shows fewer Ki67 immunoreactive cells. (**D**) Quantitative analysis of all the tumors reacted with the Ki67 antibody shows fewer immunoreactive cells in the mice treated with the *muKras* and *WT-Kras* siRNA compared to controls (*p* < 0.0001). (**E**) Kaplan Meier survival curves shows a significant survival benefit in mice treated with the NP loaded with *Kras* siRNA compared to controls and those treated with the scrambled siRNA (*p* = 0.0178). **** *p* < 0.0001.

**Table 1 ijms-24-00752-t001:** Serum chemistries in wild type C57BL/6 mice or *mutant Kras* mice (KC).

Mouse Strain	C57BL/6	C57BL/6	KC	KC	KC
Treatment	Control	*GAST* siRNA	Control	Scrambled siRNA	*MuKras* siRNA
Blood Test/(Range)					
Alkaline Phosphatase	56 ± 12	29 ± 5	76 ± 7	77 ± 8	74 ± 6
(9.6–218.85 U/L)					
Alanine aminotransferase	26 ± 6	13 ± 2	38 ± 9	60 ± 11	55 ±17
(15.00–80.10 U/L)					
Aspartate aminotransferase	159 ± 24	167 ± 52	156.75	122 ± 39	165 ± 22
(33.95–268.47 U/L)					
Albumin	3.2 ± 0.1	2.5 ± 0.2	3.3 ± 0.1	3.3 ± 0.2	3.2 ± 0.2
(1.92–4.11 g/dL)					
Bilirubin	0.3 ± 0.04	0.3 ± 0.05	0.33 ± 0.08	0.35 ± 0.09	0.3 ± 0.04
(0.17–0.53 mg/dL)					
Calcium	10.9 ± 0.6	10.9 ± 0.3	11.2 ± 0.2	11.5 ± 0.2	11.4 ± 0.4
(8.28–12.27 mg/dL)					
Creatinine	0.4 ± 0.03	0.3 ± 0.07	0.4 ± 0.04	0.4 ± 0.01	0.4 ± 0.04
(0.12–0.43 mg/dL)					
BUN	24 ± 2.7	30 ± 2.6	25 ± 2.1	25 ± 4.1	25 ± 2.1
(9.42–31.53 mg/dL)					
Amylase	529 ± 34	533 ± 199	853 ± 126	1274 ± 503	1254 ± 323
(351-1563 U/L)					
Sodium	141 ± 0.6	141 ± 0.4	141 ± 0.6	142 ± 0.8	143 ± 0.6
(124.8–160.3 mmol/L)					
Potassium	4 ± 0.06	4 ± 0.06	4.1 ± 0.04	4.2 ± 0.09	4.2 ± 0.04
(2.17–8.18 mmol/L)					
Chloride	106 ± 1.6	105 ± 2.3	109 ± 1.0	106 ± 2.6	108 ± 0.8
(99.96–121.75 mmol/L)					

**Table 2 ijms-24-00752-t002:** Experimental design of five separate in vivo models.

Mouse Strain	Experiment	Treatments (N)	Objective	Sex
Athymic nude	Orthotopic Human AsPC-1	PBS control (7),	Delivering 2 payloads. Measuring metastases	Female
	10^6^ cells/mouse	NPs: Scrambled (7), *GAST*, (8)	Dose 240 nM TIW	
		*muKRAS* (8), Combination siRNA (8)		
Athymic nude	Orthotopic Human PANC-1 × 10^6^ cells/mouse	PBS control (8)	Survival study, large tumor burden &	Female
		NP: *GAST* siRNA (10)	Metastases; Dose 480 nM TIW	
C57BL/6	Orthotopic mouse mT3	PBS Control (10), NPs: Scrambled (10), Combination *muKras* and *WT- Kras* siRNA (10)	Ki67 proliferation	Female
	100,000 cells/mouse		Survival	
			Deliver 2 payloads	
			Dose 240 nM TIW	
Transgenic	*P48-Cre/*LSL-*Kras*^G12D/+^	Control (no RX, 8), NPs: Scrambled siRNA (8), *muKras* (8)	Prevent PanINs	Males and females
			Dose 480 nM BIW, long-term safety	
C57BL/6	Subcutaneous mT3	PBS Control (4)	Safety short term and off target toxicity	Males and females
		Scrambled siRNA NP (4)		

## Data Availability

The data generated in this study are available upon request from the corresponding author.

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
