# Peer review of "Target-Specific Nanoparticle Polyplex Down-Regulates Mutant Kras to Prevent Pancreatic Carcinogenesis and Halt Tumor Progression"

_ijms, 2023, doi:10.3390/ijms24010752_

Round 1

Reviewer 1 Report

The manuscript is full of typos and I suggest a thorough re-reading.

Regarding the experiments, I think some minor revisions are required:

1.     1. The authors has performed Masson Trichrome staining to show fibrosis, but I think this experiment should be accompanied by Sirius Red staining, which specifically stains collagen fibers.

2.     Figure 3B shows the gastrin mRNA fold change in tumors of gastrin siRNA NP-treated mice compared to controls. The authors should quantify the protein levels of gastrin in the formed tumors by western blotting.

Author Response

Reviewer 1

  1. The authors has performed Masson Trichrome staining to show fibrosis, but I think this experiment should be accompanied by Sirius Red staining, which specifically stains collagen fibers.

Response: We have now also done Sirius Red stain on the pancreas from the Kras mice and have included representative photos and densitometry analysis. These results support that the nanoparticle that binds to the CCK-BR decreased fibrosis and the collagenous component of the fibrosis.

  1. Figure 3B shows the gastrin mRNA fold change in tumors of gastrin siRNA NP-treated mice compared to controls. The authors should quantify the protein levels of gastrin in the formed tumors by western blotting.

Response: Gastrin-17 peptide is very small (<25 kD) so Western blots are usually difficult to perform. The majority measure gastrin today with ELISA for quantification. We therefore homogenized the control and gastrin-siRNA treated AsPC-1 tumors and performed and ELISA confirming the peptide was indeed significantly decreased in the tumors. This data is now added to the revised manuscript.

Thank you for your suggestions. We believe with these revisions the manuscript is significantly improved.

Reviewer 2 Report

The article entitled:"Target-specific nanoparticle down-regulates mutant kras to prevent pancreatic carcinogenesis and tumor progression", by Smith et al., is overall well written, introduced and detailed. Here, new interesting approach against pancreatic cancer is described based on nanoparticles loaded with siRNA to mutant KRAS targeted against cholecystokinin-B receptor.

Please find below some points to improve the scientific significance of the manuscript:

1. Authors should review the whole manuscript to unify KRAS in upper case an italics.

2. Ref [3] is to old, I strongly recommend substitute by a newest one.

3. When describing treatment options (Lines from 43-45), Authors did not mention any of the regimens. Please include references to PRODIGE and APACT trials.

4. I understand that Ref[19] was your previous research where you engineered and designed the particles. However, I would appreciate here a short explanation of  how nanoparticles target the cholecystokinin- B receptor selectively.

5.- Why is "prevent" underlined  in line 85?

6.- Abbreviations must be define at first use e.g.: Gastrin (GAST), muKras (mutant KRAS), etc.

 7. Images from Figure 1A, 1B 1C, 2A, 2B, 2C, 3D, 3E, 3F, 5A, 5B, and 5C should contain in a corner a 40X or 100X photograph to appreciate individual cells with their nuclei as in figure 2E, 2G or 2F.

 8. Lines 168-169, Authors state  "M2 polarized macrophages were quantified and significantly less in the pancreas of mice treated with the targeted NP"; therefore, include the p-value of the statistical analysis.

9. When Authors point "arginase positive  M2- macrophagues", how can they distinguish between M2-macrophagues and neutrophils that also present high content in arginase?

10. In the Result section: "2.5. CCK-BR Targeted NPs Exhibit Broad Safety Profile and Lack off Target Toxicity " I expected other quantifications like inflammatory cytokines or white blood cells count specially neutrophils to evaluate neutropenia as a major toxicity event after chemotherapy.

11. In materials and methods please include the number of cells (AsPC-1 and PANC-1) injected into the mice.

 12. A complete and detailed section must be included in materials and methods about statistical analysis where authors must justify why and no other test is applied in each case.

13. Conclusions are supported by results; however, since authors used transient siRNA the effect will never reach complete remission in patients and other strategies should be evaluated like shRNA. This should be included in conclusion.

Author Response

Reviewer 2

Comments and Suggestions for Authors

The article entitled:"Target-specific nanoparticle down-regulates mutant kras to prevent pancreatic carcinogenesis and tumor progression", by Smith et al., is overall well written, introduced and detailed. Here, new interesting approach against pancreatic cancer is described based on nanoparticles loaded with siRNA to mutant KRAS targeted against cholecystokinin-B receptor.

Please find below some points to improve the scientific significance of the manuscript:

  1. Authors should review the whole manuscript to unify KRASin upper case and italics.

Response: Humans, non-human primates, chickens, and domestic species: Gene symbols contain three to six italicized characters that are all in upper-case (e.g., AFP). Protein symbols are identical to their corresponding gene symbols except that they are not italicized (e.g., AFP).

Mice and rats: Gene symbols are italicized, with only the first letter in upper-case (e.g., Gfap). Protein symbols are not italicized, and all letters are in upper-case (e.g., GFAP).

Therefore, where we referred to Human genes the GAST and KRAS are capitalized and italicized. When we referred to mouse, only the first letter is capitalized Kras

  1. Ref [3] is to old, I strongly recommend substitute by a newest one.

Response: The old Ref 3 was replaced with the following latest review from 2020: Mizrahi JD, Surana R, Valle JW, Shroff RT. Pancreatic cancer. Lancet. 2020 Jun 27;395(10242):2008-2020. doi: 10.1016/S0140-6736(20)30974-0. PMID: 32593337.

  1. When describing treatment options (Lines from 43-45), Authors did not mention any of the regimens. Please include references to PRODIGE and APACT trials.

Response: These references do not refer to those with advanced pancreatic cancer (as in our Introduction) but to subjects that underwent resection and had no visible disease.  In response to this reviewer, we have added a line in the introduction referring to these studies in resected subjects.

  1. I understand that Ref [19] was your previous research where you engineered and designed the particles. However, I would appreciate here a short explanation of  how nanoparticles target the cholecystokinin- B receptor selectively.

Response: As described in the method section (4.2) “To render the NP target-specific for the cholecystokinin-B receptor (CCK-BR) we conjugated maleimide functionalized Gastrin-10 peptide (Glu-Glu-Glu-Ala-Tyr-Gly-Trip-Met-Asp-Phe-NH2, MW 1426.48 g/mol) (custom synthesis by GenScript USA Inc., Piscataway, NJ, USA) to SH-PEG-PLL via a Michael addition reaction.”

5.- Why is "prevent" underlined  in line 85?

Response: This was for emphasis, but we have removed the underline.

6.- Abbreviations must be define at first use e.g.: Gastrin (GAST), muKras (mutant KRAS), etc.

  1. Images from Figure 1A, 1B 1C, 2A, 2B, 2C, 3D, 3E, 3F, 5A, 5B, and 5C should contain in a corner a 40X or 100X photograph to appreciate individual cells with their nuclei as in figure 2E, 2G or 2F.

Response: Since eye pieces vary in microscopes the correct representation of the image size is used with the bar scale as shown in all these images. For the slides scanned with the Aperio microscope the computerized software places the scale bar according to the exact magnification. 500um is usually 4X, 200um is 10X, and 100um is 20X.  Some of our images were taken at the best resolution for that figure on the computerized Aperio microscope (300um=  8.4X and 400um=5.4X). We have added the magnification to the figure legends.

  1. Lines 168-169, Authors state  "M2 polarized macrophages were quantified and significantly less in the pancreas of mice treated with the targeted NP"; therefore, include the p-value of the statistical analysis.

Response: The P value was on the Figure that was referred to; however, we have now added the P value also to the text.

  1. When Authors point "arginase positive  M2- macrophages", how can they distinguish between M2-macrophagues and neutrophils that also present high content in arginase?

Response: In order to respond to this comment and confirm the macrophages were changed with the therapy, we re-sectioned and performed immunohistochemistry on the slides with F4/80 to show the arginase staining was from macrophages. These figures and analysis are now added to the revised manuscript.

  1. In the Result section: "2.5. CCK-BR Targeted NPs Exhibit Broad Safety Profile and Lack off Target Toxicity " I expected other quantifications like inflammatory cytokines or white blood cells count specially neutrophils to evaluate neutropenia as a major toxicity event after chemotherapy.

Response: We attempted to perform a cytokine analysis but there was not enough serum left from the mice after performing the serum chemistries.

  1. In materials and methods please include the number of cells (AsPC-1 and PANC-1) injected into the mice.

Response: This information was provided in the Methods section in the Table summarizing all 5 in vivo experiments.

  1. A complete and detailed section must be included in materials and methods about statistical analysis where authors must justify why and no other test is applied in each case.

Response; Statistical section is added.

  1. Conclusions are supported by results; however, since authors used transient siRNA the effect will never reach complete remission in patients and other strategies should be evaluated like shRNA. This should be included in conclusion.

Response: We disagree with this comment. Even chemotherapy has to be given repeatedly due to the T1/2. Stable knockdown with shRNA requires a plasmid and these agents have not yet been approved by the FDA. Also, we specifically state that therapy today for pancreatic cancer does not use monotherapy. Therefore since we found an agent that decreases metastases without toxicity could easily be administered in combination with other regimens. Since our work in the mutant Kras mice showed we could also prevent PanIN progression, using this therapy could be a potentially less toxic therapy for those with resected disease such as referred to above in the PRODIGE and APACT trials.

Thank you for your comments. We believe these revisions significantly improve the manuscript.

Round 2

Reviewer 2 Report

Thanks for your comments.